Spintharus flavidus in the Caribbean—a 30 million year biogeographical history and radiation of a ‘widespread species’

Dziki Austin 1
Binford Greta J. 2
Coddington Jonathan A. 3
Agnarsson Ingi 1 3 iagnarsson@gmail.com
1 Department of Biology, University of Vermont , Burlington, VT , USA
2 Department of Biology, Lewis and Clark College , Portland, OR , USA
3 Department of Entomology, National Museum of Natural History, Smithsonian Institution , Washington, DC , USA
Grandcolas Philippe
Electronic publication date: 2015 Nov 19
Publication date: 2015
Volume: 3
Electronic Location ID: e1422
Received 2015 Aug 28; Accepted 2015 Nov 2
Copyright: © 2015 Dziki et al.
Copyright year: 2015
Copyright holder: Dziki et al.
License: This is an open access article distributed under the terms of the Creative Commons Attribution License, which permits unrestricted use, distribution, reproduction and adaptation in any medium and for any purpose provided that it is properly attributed. For attribution, the original author(s), title, publication source (PeerJ) and either DOI or URL of the article must be cited.
License URL: https://creativecommons.org/licenses/by/4.0/

Keywords: Spintharus, GAARLandia, Theridiidae, Evolution, Caribbean, Adaptive radiation, Dispersal, Zoology, Arachnology

Funding: National Science Foundation DEB-1314749 DEB-1050187-1050253 NSF DBI-1349205 Smithsonian Institution 2013 SI Barcode Network UVM APLE grant Funding for this work comes from National Science Foundation (DEB-1314749 and DEB-1050187-1050253) to I Agnarsson and G Binford and NSF (DBI-1349205) to D Barrington, I Agnarsson and CW Kilpatrick. Additional funds came from the Smithsonian Institution 2013 SI Barcode Network to JA Coddington and I Agnarsson. Development of this project was further supported by a UVM APLE grant to A Dziki. The funders had no role in study design, data collection and analysis, decision to publish, or preparation of the manuscript.

==============================
The Caribbean island biota is characterized by high levels of endemism, the result of an interplay between colonization opportunities on islands and effective oceanic barriers among them. A relatively small percentage of the biota is represented by ‘widespread species,’ presumably taxa for which oceanic barriers are ineffective. Few studies have explored in detail the genetic structure of widespread Caribbean taxa. The cobweb spider Spintharus flavidus Hentz, 1850 (Theridiidae) is one of two described Spintharus species and is unique in being widely distributed from northern N. America to Brazil and throughout the Caribbean. As a taxonomic hypothesis, Spintharus “flavidus” predicts maintenance of gene flow among Caribbean islands, a prediction that seems contradicted by known S. flavidus biology, which suggests limited dispersal ability. As part of an extensive survey of Caribbean arachnids (project CarBio), we conducted the first molecular phylogenetic analysis of S. flavidus with the primary goal of testing the ‘widespread species’ hypothesis. Our results, while limited to three molecular loci, reject the hypothesis of a single widespread species. Instead this lineage seems to represent a radiation with at least 16 species in the Caribbean region. Nearly all are short range endemics with several distinct mainland groups and others are single island endemics. While limited taxon sampling, with a single specimen from S. America, constrains what we can infer about the biogeographical history of the lineage, clear patterns still emerge. Consistent with limited overwater dispersal, we find evidence for a single colonization of the Caribbean about 30 million years ago, coinciding with the timing of the GAARLandia landbridge hypothesis. In sum, S. “flavidus” is not a single species capable of frequent overwater dispersal, but rather a 30 my old radiation of single island endemics that provides preliminary support for a complex and contested geological hypothesis.

Introduction

Archipelagos represent unique conditions to study gene flow and diversification (Agnarsson & Kuntner, 2012; Gillespie & Roderick, 2002; Losos & Ricklefs, 2010; Ricklefs & Bermingham, 2008; Warren et al., 2015). Islands are situated within a matrix of oceanic barriers that restrict gene flow in proportion to the geographic isolation of an island and the dispersal ability of a given taxon (Agnarsson, Cheng & Kuntner, 2014; Claramunt et al., 2012; Diamond, Gilpin & Mayr, 1976; Warren et al., 2015). They also provide opportunity for adaptive radiations within islands, particularly for dispersal-limited taxa.

Among archipelagos, the Caribbean is particularly rich as an arena for biogeographic analyses. The region is composed of a range of old continental fragments (Greater Antilles, ∼40 million years (my)) and relatively recent volcanic islands (Lesser Antilles, <10 my) (Iturralde-Vinent & MacPhee, 1999; Iturralde-Vinent, 2006), and features rich biodiversity and high levels of endemism (Hedges, 1996a; Hedges & Heinicke, 2007; Losos & DeQueiroz, 1997; Myers et al., 2000; Ricklefs & Bermingham, 2008). The proximity of these islands to ancient continents (N. and S. America), creates potential for a dynamic interchange of taxa between continents and islands (Bellemain & Ricklefs, 2008; Heaney, 2007). The geological history of the islands has created unique conditions for colonization and speciation. Over the last 40 my, the Greater Antilles landmasses have emerged, moved, sometimes amalgamated, and perhaps, connected to South America via a former land bridge; the Greater Antilles and Aves Ridge Land Bridge hypothesis (GAARLandia) (Iturralde-Vinent & MacPhee, 1999). Ricklefs & Bermingham (2008) portrayed the Caribbean as “a laboratory of biogeography and evolution” ideally suited to study replicate patterns of allopatric speciation and evolutionary radiations.

Arthropods can strongly test biogeographical patterns (Gillespie & Roderick, 2002). They can be abundant enough to be easily sampled without deleterious population effects, and they have short generation times, compared to vertebrates and many plants. Short generation times and large brood sizes allow them to evolve and diversify quickly. Like vertebrates and plants, arthropods span the spectrum of dispersal abilities from extremely poor to excellent dispersers. Widespread species may be regarded as good dispersers and for a ‘naturally’ widespread species to persist in the Caribbean, gene flow among islands must be sufficient. Thus, for lineages with relatively poor dispersal ability, widespread species are improbable taxonomic hypotheses.

The taxonomic hypothesis Spintharus flavidus circumscribes a widespread spider species found throughout the Caribbean and from northern N. America to Brazil (Levi, 1954; Levi, 1963). This species has been documented ballooning (aerial dispersal on silk threads (Bell et al., 2005)) and might be expected to maintain gene flow across oceanic barriers. However, the single ballooning record was a short distance dispersal, and other information on the distribution and biology of Spintharus suggest long distance dispersal is rare. Spintharus have a somewhat cryptic habitat, often found in leaf litter or the undersides of low-level leaves. The genus is old; its sister lineage Episinus occurs in 45 my Baltic amber (Wunderlich, 2008; Wunderlich, 2012; Wunderlich, 2015), but, unlike Episinus, Spintharus is restricted to the Americas. Spintharus includes just two species, the putatively widespread S. flavidus and S. gracilis restricted to Brazil (Levi, 1954; Levi, 1963). A third species S. argenteus (Dyal 1935) from Pakistan is clearly misplaced in the genus based on its original description (Levi, 1954), it’s proper placement is unclear but it may be a tetragnathid (I Agnarsson, pers. obs., 2015). Spintharus flavidus varies greatly in color and genitalic morphology—usually species-specific in spiders—thus hinting at greater species diversity. Levi (1954) and Levi (1963), however, did not see clear geographical patterns to this variation. Specimens from a single locality differed so continuously in color and genitalia that he believed it to be one widespread and variable species.

Preliminary analyses of the COI gene indicated high levels of molecular diversity within S. flavidus and contested the widespread species hypothesis. Here, we use molecular phylogenetics to study patterns of diversification in a ‘widespread’ spider species, specifically to test the hypothesis that it represents single species capable of frequent dispersal across moderate stretches of ocean. While our taxon sampling is not designed for detailed biogeographical analyses, samples from the continental landmasses (N., C. and S. America) permit preliminary evaluation of the route, number, and timing of colonization events. We also use our data to propose an initial biogeographical hypothesis for Spintharus that can readily be tested as more data accumulate, especially from South America.

Materials and Methods

Specimen sampling and DNA extraction and assembly

The CarBio team (www.islandbiogeography.org) collected specimens from Cuba, the Dominican Republic, Puerto Rico, Jamaica, the Lesser Antilles, Florida, South Carolina, Costa Rica, Mexico, and Columbia between 2011 and 2015. All specimens were collected under appropriate permits: Puerto Rico, DRNA: 2011-IC-035 (O-VS-PVS15-SJ-00474-08042011); Jamaica, NEPA, reference number #18/27; USA, USDI National Park Service, EVER-2013-SCI-0028; Costa Rica, SINAC, pasaporte científico no. 05933, resolución no. 019-2013-SINAC; Cuba, Departamento de Recursos Naturales, PE 2012/05, 2012003 and 2012001; Dominican Republic, Ministerio de Medio Ambiente y Recursos Naturales, no 0577, Mexico, SEMARNAT scientific collector permit FAUT-0175 issued to Dr. Oscar Federico Francke Ballve, Oficio no. SGPA/DGVS/10102/13; Colombia, Authoridad Nacional de Licencias Ambientales, 18.497.666 issued to Alexander Gómez Mejía; Saba, The Executive Council of the Public Entity Saba, no 112/2013; Martinique, Ministère de L’Écologie, du Développement Durable, et de L‘Énergie; Nevis, Nevis Historical & Conservation Society, no F001; Barbados, Ministry of Environment and Drainage, no 8434/56/1 Vol. II. We used standard protocols for aerial search, beating, sifting, and cryptic methods (Coddington et al., 2009; Coddington et al., 1991). Spiders were immediately fixed in 95% ethanol and stored at −20 °C (UVM Natural History Museum). Phenotype vouchers will also be deposited at the USNM (Smithsonian Institution).

We chose 195 individuals from the field samples for molecular analyses, representing each collecting locality with several specimens (targeting 4–5), when available (Fig. 1). DNA was extracted from 1 to 4 legs from each individual and isolated using a Qiagen DNeasy Tissue kit using the kit protocol (Qiagen, Valencia, CA, USA). DNA from some specimens was isolated from leg samples at the Smithsonian Institute (SI) in Washington, D.C. using an Autogenprep965 for an automated phenol chloroform extraction. We initially sequenced a fragment of the mitochondrial cytochrome c oxidase subunit 1 (COI) to establish basic patterns of phylogenetic relationships and obtain an initial estimate of diversification patterns through ‘DNA barcoding’ (Hebert et al., 2003). Upon discovering deep divergences among isolated ‘populations’ with COI, we additionally sequenced the mitochondrial ribosomal 16S rRNA (16S), and the nuclear Internal Transcribed Spacer unit 2 (ITS2) for selected exemplars from each well supported regional clade indicated by COI. These molecular markers have proven successful in similar phylogenetic studies of spiders ranging from low taxonomic levels to divergences as deep as the age of the Caribbean (Agnarsson, 2010; Agnarsson & Rayor, 2013; Kuntner & Agnarsson, 2011a; Kuntner & Agnarsson, 2011b). To amplify the COI, 16S, and ITS2, we used LCOI-1490 & HCOI-2198, 16S A & 16S B, and ITS 4 & ITS 5.8 primers respectively (Folmer et al., 1994; Simon et al., 1994; White et al., 1990). Standard PCR protocols were used as described in (Agnarsson, 2010; Agnarsson, Maddison & Aviles, 2007). The PCR products were purified using Exosap kits and purified PCR products sequenced at the University of Arizona, Beckman Genomics, or the Smithsonian Institution. All sequences were deposited in GenBank (accession numbers not yet available). COI, 16S, and ITS2 sequences from Anelosimus and Episinus species, the latter a closer relative of Spintharus (Agnarsson, 2004; Arnedo et al., 2004) were downloaded from GenBank and used as outgroups along with a Chrysso specimen we sequenced here.

Figure 1 Phylogenetic tree.

Results of a Bayesian analysis of the three concatenated loci summarized in terms of region and clade uniqueness. Outgroups are omitted for clarity. Color coded and numbered clades represent our initial species hypotheses based on this tree and barcoding gaps. Encircled numbers on nodes refer to our ‘conservative’ estimtes of actual species richness (see Table 1). Numbers below nodes are posterior probability values, bold indicate clades recovered in the Bayesian analysis of the small nuDNA dataset alone. Results from maximum likelihood analyses of the same dataset are largely congruent, numbers above clades are ML bootstraps. For details of specimens in each clade see Supplemental Information 1 and Table S1. Inset map shows collecting sites for this study.

The chromatographs were interpreted with Phred 45 and Phrap (Green, 2009; Green & Ewing, 2002) using the Chromaseq module (Maddison & Maddison, 2011a) implemented in the program Mesquite (Maddison & Maddison, 2011b) and edited by hand. The alignments for the COI sequences were trivial with no implied indels, and were done in Mesquite through ClustalW. The 16S and ITS2 sequences were aligned using the program MAFFT (Katoh, 2013) through the online server portal http://mafft.cbrc.jp/alignment/server using default settings other than setting the tree building number and maxiterate to the maximum.

Phylogenetic and biogeographical analyses

The aligned sequences for COI, 16S, and ITS2 were tested for the best fitting substitution model using the program Jmodeltest 2.1.7 (Darriba et al., 2012). The best models for each gene, among the 24 models available in MrBayes, were GTR + G for 16S and ITS2 and GTR + I + G for COI.

We used MrBayes V3.2.3 on XSDE (Ronquist et al., 2012) through the online portal CIPRES (Miller et al., 2015), to run a Bayesian analysis for mtDNA (COI plus 16S) and nuDNA (ITS2) separately and for the concatenated three loci. We used Mesquite to concatenate loci and to partition the analysis by locus. The Bayesian analyses ran Metropolis coupled Markov chain Monte Carlo (MC3) (for 50,000,000 generations), sampling every 1,000 generations. We used Tracer (Drummond & Rambaut, 2007), to insure proper convergence of runs, and sufficient sampling of priors. An analysis partitioned additionally by codons yielded nearly identical results. Maximum likelihood (ML) searches were done in Garli 2.0, (Zwickl, 2006) and repeated 100 times and the tree maximizing likelihood of the data was preferred.

Analyses of divergence times were done in BEAST 1.8 (Drummond et al., 2012). We pruned the matrix to include 2–3 exemplars with minimum missing data from each major clade, and constrained the monophyly of Spintharus and that of Dominican Spintharus, because prior analyses in MrBayes and ML analyses justified their monophyly. We employed GTR + G + I model for the concatenated matrix with a Yule process tree prior, and a UPGMA starting tree. We used a burnin of 5000 with maximum clade credibility tree target. We also ran separate analysis using a birth–death model (see Condamine et al., 2015 for detail), an analysis partitioned by gene, a dating analysis of COI alone calibrated by estimated rates of evolution for that gene (Bidegaray-Batista & Arnedo, 2011), and a coalescent gene tree-species tree analysis in *BEAST. Two chains of 100 million generations were run and, convergence and correct mixing of the chains were monitored using Tracer 1.5. Priors were set to default other than detailed below. We estimated node ages using a relaxed exponential clock calibrated with a Dominican amber fossil of Spintharus that is dated to about 15–20 mya (Wunderlich, 1988). This sets the minimum age of the genus, but more importantly, the colonization of Hispaniola at 15 my ago and was implemented using exponential priors on both nodes with a mean of 5 and offset of 15 my—spanning about 15–35 my. This represents a time span from the estimated age of the fossil until prior to GAARlandia and close to the maximum age of extant Caribbean lineages (Iturralde-Vinent and 1999). Many fossils of the closely related Spintharinae genus Episinus s. l. occur in Dominican and Baltic amber (Wunderlich, 1988; Wunderlich, 2008), the latter dating to approximately 44 my. Hence, we constrained the root of Spintharinae (Spintharus plus Episinus) with an exponential prior with an offset of 44 my and mean of 15, spanning approximately 40–100 my, to the approximate origin of Theridiidae (Bond et al., 2014; Liu et al., 2015). A recent study on Theridiidae estimates the origin of Spintharinae between 55–45 mya (Liu et al., 2015).

We inferred ancestral ranges using RASP 3.2 (Yu et al., 2015) inputting the preferred Bayesian tree and a set of 100 post-burnin trees. We defined areas as each of the Caribbean islands, and S. America, and N. America (including USA, Costa Rica, and Mexico). S-DIVA, S-DEC and Bayesian Binary ancestral area analyses were run limiting areas to two, as all putative species level clades are restricted to one area, and without dispersal constraints. We exported results as tables and graphics and the latter we touched up in Adobe Illustrator.

Species delimitation, distribution, and photo-documentation

We calculated distances among clades suggested by the barcoding analysis of the COI data using MEGA6 (Tables S2 and 1). The phylogenetic results and genetic distance measures (approximately 5–10+ times greater distance among than within putative species), plus locality information (regionally monophyletic groups) provide initial species hypotheses. Various species delimitation methods were then used to help estimate number of species in this radiation using COI or the three loci dataset/tree depending on the method. We used the species delimitation plugin in Geneious 8.1.5 (Kearse et al., 2012; Masters, Fan & Ross, 2011) to estimate species limits under Rosenberg’s reciprocal monophyly P(AB) (Rosenberg, 2007) and Rodrigo’s P(RD) method (Rodrigo et al., 2008). We also estimated the probability of population identification of a hypothetical sample based on the groups being tested (P ID(Strict) and P ID (Liberal)). The genealogical sorting index (gsi) statistic (Cummings, Neel & Shaw, 2008) was calculated using the gsi webserver (http://genealogicalsorting.org) on the estimated tree and an assignment file that contained the same user specified groups identified in the Geneious plugin. Finally we used a single locus Bayesian implementation (bPTP) of the Poisson tree processes model (Zhang et al., 2013) to infer putative species boundaries on a given single locus phylogenetic input tree available on the webserver: http://species.hits.org/ptp/. The analysis was run as a rooted tree from the MrBayes analysis, with outgroups removed for 100,000 generations with 10% burnin removed.

Table 1 Results of species delimitation analyses.

Summary of species delimitation. Species hypotheses (first column) represent colored and numbered clades on Fig. 1. The various measures of distance and isolation and exclusivity metrics of these clades follow including: distance (D), the probability of population identification of a hypothetical sample based on the groups being tested (P ID(Strict) and P ID (Liberal)), Rosenberg’s reciprocal monophyly (P(AB)), the genealogical sorting index (gsi), and a single locus Bayesian implementation of the Poisson tree processes model (bPTP). Sp congru. refers to species hypothesis that are congruent with all methods, and Sp cons. is our conservative estimate of actual species richness based on agreement among all methods and >2% mtDNA sequence divergence.

Sp Hyp.	Mono	D Intra	D Inter	Dtra/Dter	P ID(Strict)	P ID(Liberal)	P(AB)	GSI	bPTP	Sp congru.	Sp cons.	
Mex 1	Yes	0.008	0.080	0.1	0.66 (0.49, 0.84)	0.90 (0.75, 1.0)	3.10E−04	1	Y	1	1	
USA 1	Yes	n/a	0.080	n/a	n/a	0.96 (0.83, 1.0)	0.02	0.61	N	2	2	
USA 3	Yes	0.001	0.025	0.04	0.71 (0.54, 0.89)	0.94 (0.80, 1.0)	0.17	1	N			
USA 2	No	0.005	0.025	0.2	0.33 (0.22, 0.44)	0.68 (0.61, 0.74)	NA	0.83	N			
Mex 2	Yes	0.002	0.083	0.02	0.58 (0.43, 0.73)	0.97 (0.82, 1.0)	1.60E−05	1	Y	3	3	
Jam	Yes	0.0004	0.067	0.005	0.76 (0.58, 0.93)	0.98 (0.84, 1.0)	4.20E−04	1	Y	4	4	
Grenada	Yes	0.0036	0.026	0.14	0.52 (0.36, 0.67)	0.89 (0.74, 1.0)	0.05	1	Y	5	5	
St Lucia	Yes	0.000	0.026	0	0.66 (0.48, 0.84)	0.90 (0.75, 1.0)	0.05	1	Y	6		
St Kitts	Yes	0.002	0.015	0.13	0.58 (0.40, 0.75)	0.82 (0.68, 0.97)	0.02	0.83	Y	7	6	
PR	Yes	0.0006	0.015	0.04	0.65 (0.47, 0.83)	0.89 (0.74, 1.0)	0.02	1	Y			
Dom	Yes	n/a	0.062	n/a	n/a	0.96 (0.83, 1.0)	0.1	1	Y	8	7	
CU1	Yes	0.0005	0.041	0.012	0.77 (0.59, 0.94)	0.99 (0.84, 1.0)	5.80E−04	1	Y	9	8	
CU2	Yes	0	0.027	0	0.75 (0.57, 0.92)	0.97 (0.83, 1.0)	1.85E−03	1	Y	10	9	
CU3	Yes	0.002	0.027	0.07	0.81 (0.70, 0.91)	0.93 (0.86, 0.99)	1.85E−03	1	Y	11	10	
DR1	Yes	0.001	0.032	0.03	0.67 (0.49, 0.85)	0.90 (0.76, 1.0)	1.80E−04	1	Y	12	11	
DR2	Yes	0.005	0.026	0.19	0.88 (0.77, 0.99)	0.96 (0.90, 1.0)	4.40E−05	1	Y	13	12	
DR3	Yes	0.002	0.026	0.07	0.65 (0.48, 0.83)	0.89 (0.75, 1.0)	0.01	1	Y	14	13	
DR4	Yes	0.003	0.031	0.01	0.75 (0.60, 0.89)	0.94 (0.83, 1.0)	0.01	0.83	Y	15		
CU4	Yes	0.002	0.010	0.2	0.73 (0.61, 0.86)	0.93 (0.83, 1.0)	1.06E−03	1	Y	16	14	
CU5	Yes	0.003	0.010	0.3	0.61 (0.47, 0.75)	0.87 (0.76, 0.98)	1.06E−03	0.83	Y			
CU6	Yes	0	0.024	0	0.73 (0.56, 0.91)	0.96 (0.81, 1.0)	0.01	0.92	Y	17	15	
CU7	Yes	0.003	0.026	0.2	0.74 (0.59, 0.88)	0.93 (0.83, 1.0)	0.01	1	Y	18	16	

For each putative molecular species-level clade, representatives from all localities were chosen for taxonomic photography. The spiders were positioned in Germ-X hand sanitizer (65% ethanol) and covered in 95% ethanol. The photographs were taken with the Visionary Digital BK Laboratory System, using a Canon 5D camera, a 65 mm macro zoom lens. Photo stacks of 30–50 slices were then compiled using the program Helicon Focus 5.3. The image was then edited in Photoshop CS6 to balance light quality, adjust for brightness, remove background blemishes, and provide a scale.

We used the online program GPS Visualizer (http://www.gpsvisualizer.com) to plot localities (Fig. 1).

Results

Specimen sampling and DNA extraction and assembly

Of the 195 individuals chosen for DNA work, 186 yielded quality DNA and 175 were successfully amplified for COI. The subset of 186 taxa that was chosen for additional sequencing yielded 180 16S sequences and 79 ITS2 sequences, representing all major clades. In all, the concatenated matrix contains 1,572 nucleotides of which 668 are COI, 682 are 16S, and 312 ITS2.

Phylogenetics

Bayesian analyses reached convergence and appropriate ESS as determined in Tracer. The bayesian inferences of the concatenated COI, 16s, and ITS2 sequences from S. flavidus provide a robust and well-resolved phylogenetic hypothesis (Fig. 1). Maximum likelihood analyses of the concatenated matrix yield nearly identical results. Independent mDNA and nuDNA were highly congruent with one another and with the concatenated analysis. No strongly supported clades in either gene tree contradict the concatenated tree, rather areas of disagreement reflect lack of resolution in gene trees. Additional sensitivity analyses—Bayesian analyses partitioned by gene and codon done in MrBayes and in BEAST yielded similar results in terms of species relationships and colonization of the Caribbean. Moreover, a species tree analysis in *BEAST resulted in near identical relationships among putative species.

All analyses support the monophyly of Spintharus and of the Caribbean taxa. Furthermore, the N. American + Caribbean specimens form a clade and within that the specimens from the Yucatan peninsula are sister to the islands clade. Deep genetic divergences occur within the USA, especially between USA1 (from Genbank) and the remaining specimens. Within the Caribbean two main clades are supported. The first contains two Cuban and one Hispaniolan clade, of three or more species each, and the Hispaniolan clade nests among the Cuban clades. The second contains the remaining islands, Jamaica, Puerto Rico, and the Lesser Antilles. Puerto Rican specimens nest within a Lesser Antilles clade.

Separate analyses of mtDNA and nuDNA markers reveal general congruence among the independent lines of evidence. Both support the monophyly of Spintharus, N. America plus the Caribbean, Yucatan plus Caribbean and the Caribbean islands. Both recover the two Cuban clades, one sister to a Hispaniolan clade, and both nest Puerto Rico within the Lesser Antilles. Both resolve most putative monophyletic species, and all islands are monophyletic (except the two Cuban clades discussed above).

Species delimitation, distribution, and photo-documentation

Bayesian inferences of the concatenated COI, 16S, and ITS2 yield 22 distinct and well supported lineages. Most are independently recovered in the mtDNA and ITS2 datasets (Supplemental Information 1). We accepted these 22 lineages, of which 20 include multiple specimens and two single specimens as initial species hypotheses (Table 1). Most putative species lineages show genetic distances >5%, and nearly all are separated by a ‘barcoding gap’ (Table 1). Shallower divergences, between 1.9 and 4% are found between clades Cuba 1 and 2 versus Cuba 3, and between USA 1 and 2, and between St. Lucia and Grenada (Table 1). The results of various methods of species delimitation reject the single widespread species hypothesis. The bPTP analysis estimated between 17 and 31 species, including all the initial 22 species hypotheses but supported one USA species. The ML and the Bayesian tree supported 17–19 species, congruent with the minimum estimate from the bPTP analyses. Other species delimitation methods yielded similar results. Eighteen putative species had P ID (liberal) of 89 or higher, 19 had significant Rosenberg values and 18 had GSI values of >82, and 14 = 100. In general methods were congruent and supported at least the 16 putative species, circled in Fig. 1. These represent 11 of the 22 initial species hypotheses, and five species whose delimitation is broader than the original hypotheses (Table 1 and Fig. 1).

All 16 species are narrow range endemics without any range overlap suggesting allopatric speciation. All are either restricted to continents or are single island endemics, except species 7, with distinct populations on St. Kitts, Nevis and Puerto Rico, and species 5, with distinct populations on Grenada and St. Lucia. The only islands that share haplotypes are St. Kitts and Nevis, small keys separated by less than 30 km of shallow ocean. The sister clade to the St. Kitts and Nevis clade is, in contrast, Puerto Rico separated by over 250 km. The largest Greater Antilles islands, Cuba and Hispaniola, harbor minor within-island radiations resulting in multiple species-level clades.

Biogeographical patterns

The dating analysis in BEAST suggests that the N. American plus Caribbean clade diverged from the sister clade represented by the Colombian taxon, between 37 and 29 mya, a time window consistent with GAARlandia (Fig. 2). The Caribbean and the N. American clades diverged between 27 and 29 my ago. Divergences corresponding to Greater Antilles island clades are old (>20 my ago), except Puerto Rico that appears to contain a recent lineage. Some Lesser Antilles islands contain lineages estimated to be older than the currently hypothesized age of the islands. Sensitivity analyses including analysis partitioned by gene and an analysis using uniform rather than exponential priors on fossils showed general congruence. Age of the Caribbean colonization events ranged between about 25–38 my ago, and in all cases the age of the Lesser Antilles clade was older than any of its islands. In these analyses the estimated rate of COI substitutions ranged from 0.006 to 0.018, encompassing the rate estimated from independent studies (0.0112, see Bidegaray-Batista & Arnedo (2011) and Kuntner et al. (2013). A dating analysis based on COI alone calibrated only with rates (no fossil information) yielded younger estimates across the tree that were inconsistent with the available fossil record.

Figure 2 Dated phylogenetic tree.

Results of a dated BEAST analysis. Numbers on scale and nodes are in mya. Stars indicate calibration points of the analysis. Blue bar represents the span of the GAARLandia landbridge and the arrow points to the timing of colonizaiton of the Caribbean plus N. America. The age of the Caribbean island clade is estimated between 33 and 19 mya.

The RASP analyses of ancestral areas is consistent with the single colonization of N. America plus the Caribbean, and the single colonization of the islands though inferential power is limited due to only a single specimen from S. America. The common ancestor of the islands is reconstructed to have occurred on mainland and islands (Cuba), consistent with the GAARlandia hypothesis, and Hispaniola is reconstructed to have been colonized from Cuba. The colonization of Puerto Rico is supported as a relatively recent (∼6 my ago) event via ancestors in the Lesser Antilles.

Discussion

Island archipelagos, terrestrial habitat surrounded by aquatic barriers, have long offered unique insight into processes of diversification (Baldwin & Sanderson, 1998; Darwin, 1859; Gillespie & Roderick, 2002; Losos et al., 1998). Widespread species on archipelagos represent taxonomic hypotheses that predict ongoing gene flow. Such hypotheses are plausible for excellent dispersers but are rendered less and less probable as dispersal ability of organisms decreases, until finally ocean barriers become completely effective (Agnarsson, Cheng & Kuntner, 2014; Agnarsson & Kuntner, 2012; Claramunt et al., 2012; Diamond, Gilpin & Mayr, 1976). Our molecular analyses refute the current taxonomy of Spintharus flavidus as a single widespread species in the Caribbean. Phylogenetic and species delimitation analyses of both mitochondrial and nuclear genes, independently and combined, strongly reject the single-species hypothesis and suggest that S. flavidus is a radiation of short-range endemics (Harvey, 2002) in the Caribbean. Morphological evidence supports the multiple species hypothesis (Fig. 3) with some highly distinct forms and, though polymorphic, consistent color patterns within islands. This finding is consistent with the limited dispersal ability of this lineage as suggested by its biology and habitat. Furthermore, despite the relatively old age of the flavidus clade, it is restricted to the Americas (unlike its sister taxon Episinus), shows small scale genetic structuring of populations, and some limited evidence for long distance dispersal (colonization of the Lesser Antilles and Puerto Rico across water).

Figure 3 Biogeographical analysis.

Results of a preliminary RASP biogeographical analysis of ancestral areas under the Bayesian binary model. The results indicate colonization of the N. American + Caribbean clade from S America, and subsequently the Caribbean from N. America (Yucatan). The ancestral state for the Caribbean is a mizsture of islands and continent. Hispaniola was colonized from Cuba and the results indicate the colonization of Puerto Rico via the Lesser Antilles. Inset photographs are of adult females from the corresponding area on the cladogram, showing a part of the diversity of external morphology, especially coloration, in this clade.

Other work supports the notion that widespread Caribbean spiders may actually be single island endemics. The CarBio project (islandbiogeography.org) has found evidence of other multiple single island endemics among hypothetical ‘widespread species’ of putatively poor dispersers (e.g., McHugh et al., 2014; Esposito et al., 2015, also unpublished data for Loxosceles, Scytodes, and Argiope). In other groups of animals and plants, dispersal limited taxa also tend to form island endemics rather than widespread species (Ricklefs & Bermingham, 2008). In contrast, some other spider lineages with a long history in the Caribbean seem to be excellent dispersers and truly widespread, such as the garden spider (Argiope, S LeQuier et al., 2015, unpublished data), several tetragnathid species, and others (I Agnarsson et al., 2015, unpublished data). Dispersal ability is a key parameter in biogeography, and as the Intermediate Dispersal Model predicts (Agnarsson, Cheng & Kuntner, 2014; Claramunt et al., 2012; Diamond, Gilpin & Mayr, 1976), can have profound impact on both distribution and diversity of organisms. Mechanisms and routes of colonization in the Caribbean are diverse across different organisms (Ricklefs & Bermingham, 2008). Among relatively poor dispersers, however, evidence mounts for an important, temporary (∼34 my ago) overland dispersal route (Alonso, Crawford & Bermingham, 2012), the GAARlandia landbridge that was proposed by Manuel Iturralde-Vinent and Ross MacPhee (e.g., Iturralde-Vinent, 1998; Iturralde-Vinent & MacPhee, 1999; Iturralde-Vinent, 2006, see also Houbena et al., 2012).

However, others have criticized the GAARlandia landbridge (Ali, 2012; Hedges, 1996b; Hedges, 2006). These authors point out that the Caribbean biota is represented by a highly restricted sample of American biota that often has radiated into unoccupied niches, suggesting effective barriers, and that evidence of a massive synchronious colonization of the islands by multiple lineages of organisms at 34 mya is weak because estimates of arrival times of different lineages are rarely identical (Ali, 2012).

Nevertheless, a number of studies support “GAARlandia” because multiple monophyletic Caribbean clades approximately date to this narrow time window, such as: several spiders and scorpions (Binford et al., 2008; Crews & Gillespie, 2010; McHugh et al., 2014) (L Esposito et al., 2015, unpublished data); toads (Alonso, Crawford & Bermingham, 2012), mammals (Davalos, 2004) (but see Fabre et al., 2014); cichlid fishes (Rican et al., 2013); butterflies (Matos-Maravi et al., 2014; Peña et al., 2010), spurge plants (Euphorbiaceae) (Van Ee et al., 2008), and others (Ricklefs & Bermingham, 2008).

Although we emphasize the preliminary nature of these biogeographical analyses for Spintharus, these spiders also support colonization via GAARlandia (Figs. 2 and 3). The Caribbean Spintharus form a clade and the Caribbean plus N. America a more inclusive clade, and the estimated timing of colonization of the Caribbean plus N. America at about 32 my ago agrees with the hypothesized age of the land bridge (Fig. 2). Ancestral area reconstruction also supports a single origin on a mix of mainland and islands, consistent with GAARlandia (Fig. 3). These results are necessarily preliminary due to lack of sampling of S. America other than a specimen from Colombia. We expect that S. flavidus in S. America also will be found to represent multiple species. However, available morphological evidence suggests that S. American specimens are more similar to the sampled Colombian specimen than the Caribbean + N. American clade (Levi, 1963). Furthermore, the other known Spintharus species is restricted to S. America (although it, too, may be a species complex). With the strength of this combined evidence, further sampling of S. American taxa seems unlikely to refute the Caribbean + N. American monophyly.

The sister relationship between the Caribbean clade and specimens from the Yucatan peninsula is intriguing (Fig. 1). The Yucatan peninsula has remained the closest continental landmass to any of the Greater Antilles Islands (Cuba) ever since the GAARlandia period and could have been colonized prior to other areas in N. America. The sister relationship between N. America and the archipelago is, in turn, frequently observed in other taxa, such as other lineages of spiders (Binford et al., 2008).

Two main lineages occur in the Caribbean (Fig. 1). The first comprises Cuban and Hispaniolan taxa. These islands are not only adjacent, but stayed connected for a time even after GAARlandia broke up, and lineages on these islands are often close relatives (Iturralde-Vinent & MacPhee, 1999; Iturralde-Vinent, 2006). Furthermore, both islands include geological subunits that were separate islands during the formation of the Caribbean. Hispaniola is composed of two islands that eventually fused, and Cuba has four highland regions that were all islands at some point. Our findings agree with this geology; one Cuban lineage is sister to Hispaniolan taxa rather than the other Cuban clade, and their ages predate the Cuba–Hispaniola separation (The Cuban Hispaniola node dates to 17.8 mya).

The other main Caribbean clade comprises Jamaican, Puerto Rican, and Lesser Antillean species. Several patterns are noteworthy. Jamaica was colonized early on, consistent with its old age. However, the colonization of the Lesser Antilles is estimated at around 18–19 my ago, long before the age of existing islands (Fig. 2). Endemic island taxa can predate the islands they currently occur on (Heads, 2011). Such lineages may have occupied older, now submerged, islands or they could have gone extinct from the mainland or other older islands. The latter absences could also be false due to sparse sampling. Other taxa, like Lesser Antillean geckos dating back some 13–14 my ago, also significantly predate their islands and are thought to have colonized islands now long gone (Thorpe et al., 2008). An alternative, and perhaps simpler explanation, is error in divergence time estimation. Indeed, an analysis of divergence times relying only on a priori estimated rates of mtDNA substitution (e.g. Kuntner et al., 2013) resulted in younger estimates of all clades, but these results (not shown) were inconsistent with the fossil record. More detailed biogeographical analyses including a greater sampling of as yet unsampled areas (importantly, S. America) may improve accuracy and will serve to test the unexpectedly old age of the Lesser Antilles group.

Puerto Rico evidently was colonized relatively recently via the Lesser Antilles (6–7 my ago, Figs. 1 and 3). The results suggest ‘island hopping’ from Grenada and St. Lucia in the south, through Dominica and then St. Kitts and Nevis in the north. Puerto Rican taxa are usually related to other Greater Antillean groups. Spintharus are probably capable of overwater dispersal over short distances. Additional sampling in the Lesser Antilles may reveal more evidence for stepping stone-like colonization. Thus, we can predict the approximate phylogenetic placement of as yet unsampled areas such as Barbados, Martinique, Guadalupe, Anguilla, and the Virgin Islands, forming a grade in among the Lesser Antilles taxa sampled here.

Interestingly the southern Lesser Antilles islands in many taxa are S. American in origin, and not a part of Caribbean clades (e.g., Esposito et al., in review). Further sampling of Spintharus in Central America and the Lesser Antilles, with focus on the southernmost islands of Trinidad and Tobago as well as the neighboring regions of Venezuela and Columbia are priorities.

The implications of our findings for conservation are profound. Instead of a common, widespread species unlikely to rank highly in conservation priorities, multiple, narrowly endemic species exist that merit independent conservation evaluation and effort. As congruent evidence from multiple lineages for small endemic areas accumulates, the basic units of conservation strategy both multiply and shrink in size.

In sum, our findings reject the hypothesis of S. flavidus as a single widespread species but rather suggest it is a radiation of at least 16 short-range endemics that colonized the Caribbean over 30 my ago.

Supplemental Information

Table S1 Specimen table

List of specimens included in this study.

Click here for additional data file.

Table S2 Table S2

Genetic distances within and among the initial 22 putative Caribbean species groups, plus the Colombian specimen as determined by an analysis in MEGA6.

Click here for additional data file.

Supplemental Information 1 Data matrix

Data matrix with the three concatenated loci

Click here for additional data file.

Supplemental Information 2 Accession numbers

Click here for additional data file.

We would like to thank all the members of the CarBio team, especially those involved in expeditions in Puerto Rico (2011), the Dominican Republic (2012), Cuba (2012), the Lesser Antilles (2013) and North America (2013) including Mexico (2014). We are especially grateful to the following for help with organizing fieldwork Alexander Sanchez (Cuba), Lauren Esposito, Gabriel de los Santos, Solanlly Carrero, and Kelvin Guerrero (Dominican Republic). Oscar Francke and Alejandro Valdez (Mexico), Lauren Esposito (Jamaica and the Lesser Antilles). Our sincere thanks to all our CarBio collaborators for participation in these fieldtrips and research, including Carlos Viquez, Abel González, Giraldo Alayon, Franklyn Cala-Riquelme, Aylin Alegre, René Barba Diaz, Hanna Madden, Rodrigo Monjaraz, Nadine Duperre, Bernhard Huber, Matjaz Kuntner, and many more (see islandbiogeography.org). Many current and graduated members of the Agnarsson and the Binford labs were also instrumental in organizing and executing fieldwork including Anne McHugh, Zamira Yusseff-Vanegas, Gigi Veve, Lisa Chamberland, Federico Lopez-Osorio, Carol Yablonsky, Sarah Kechejian, Laura Caicedo-Quiroga, Jose Sanchez, Angela Alicea, Trevor Bloom, Ian Petersen, Alex Nishita, Katy Loubet-Senear, Sasha Bishop, Charlotte Francisco, Eva Ramey, Ian Voorhees, Angela Chuang, Micah Machina and many more. Maxwell Stuart helped with photography and DNA work. All material was collected under appropriate collection permits and approved guidelines. Additional logistic support was provided by Fideicomiso de Conservación de Puerto Rico, Universidad Interamericana de Puerto Rico, and Casa Verde, Maunabo. Comments from Fabian Condamine, Miquel Arnedo and Jason Ali greatly improved the manuscript.

Additional Information and Declarations

Competing Interests

Author Contributions

Field Study Permissions

DNA Deposition

The authors declare there are no competing interests.

Austin Dziki conceived and designed the experiments, performed the experiments, analyzed the data, wrote the paper, prepared figures and/or tables, reviewed drafts of the paper.

Greta J. Binford conceived and designed the experiments, contributed reagents/materials/analysis tools, wrote the paper, reviewed drafts of the paper.

Jonathan A. Coddington wrote the paper, reviewed drafts of the paper.

Ingi Agnarsson conceived and designed the experiments, analyzed the data, contributed reagents/materials/analysis tools, wrote the paper, prepared figures and/or tables, reviewed drafts of the paper.

The following information was supplied relating to field study approvals (i.e., approving body and any reference numbers):

All specimens were collected under appropriate permits: Puerto Rico, DRNA: 2011-IC-035 (O-VS-PVS15-SJ-00474-08042011); Jamaica, NEPA, reference number #18/27;

USA, USDI National Park Service, EVER-2013-SCI-0028; Costa Rica, SINAC, pasaporte científico no. 05933, resolución no. 019-2013-SINAC; Cuba, Departamento de Recursos Naturales, PE 2012/05, 2012003 and 2012001; Dominican Republic, Secretaría de Estado de Medio Ambiente y Recursos Naturales Ministerio de Medio Ambiente y Recursos Naturales, no 0577, Mexico, SEMARNAT scientific collector permit FAUT-0175 issued to Dr. Oscar Federico Francke Ballve, Oficio no. SGPA/DGVS/10102/13; Colombia, Authoridad Nacional de Licencias Ambientales, 18.497.666 issued to Alexander Gómez Mejía; Saba, The Executive Council of the Public Entity Saba, no 112/2013; Martinique, Ministère de L’Écologie, du Développement Durable, et de L‘Énergie; Nevis, Nevis Historical & Conservation Society, no F001; Barbados, Ministry of Environment and Drainage, no 8434/56/1 Vol. II.

The following information was supplied regarding the deposition of DNA sequences:

Genbank accession numbers are provided in the Supplemental Information 2 file.

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
