# Peer review of "Spintharus flavidus in the Caribbean—a 30 million year biogeographical history and radiation of a ‘widespread species’"

_PeerJ, doi:10.7717/peerj.1422_

## Round 0.1 · original submission · Major Revisions

Thank you for submitting your interesting manuscript. The three referees have found your study very interesting and valuable. They however made a series of comments that should help to improve your manuscript. I recommend that you address every comment made by the three referees, either with a modification of with a rebuttal. I found these comments very positive and very timely and I would be really glad that you try to take into account most of them and then resubmit your paper to PeerJ.

·

Basic reporting

In this paper, the authors (Dziki et al.) investigate the phylogenetic relationships as well as the divergence times of a widespread spider species in the Caribbean islands (but also spread in N. and S. Americas). The authors use a molecular dataset composed of three genes and 195 sampled taxa (not all worked for the sequencing). Based on their phylogenetic analyses they used species delimitation approaches to evaluate the hypothesis of single widespread species vs. multiple short-range endemic. They further used molecular dating analyses to study the origin of the Caribbean spider biodiversity, in relation with geological changes in the region over the last 30 Myrs. They tested the hypothesis of single vs. multiple colonization event(s) and subsequent diversification in the archipelago.

In sum, they found a well-resolved (but with mixed support depending on the gene trees) phylogeny that allows the delineation of 16 putative species. Clearly the speciation mode of the species is allopatric. The dating estimated the origin of the Spintharus radiation at ca. 30 Myrs ago. The time-calibrated tree also shows that the Caribbean fauna forms a single monophyletic group that colonized the Caribbean islands only once via the GAARlandia land bridge. As acknowledged by the authors the latter has still to be tested with more taxon sampling from S. America.

I enjoy reading the paper, and I have made some comments to improve the paper. I think the authors did not explain enough how the analyses have been conducted (the Material and Methods suffer a lack of details) and sometimes the analyses may be a bit unclear. The results are interesting for the field of spider phylogeny and taxonomy, and island biogeography. The paper is well written but can be improved as well.

This paper is a taxon-oriented paper, but taking a biological model as an example to test the hypothesis of widespread species in an island system. I think the paper will be of interest to specialists of the spiders but also in general given the question addressed on the dispersal ability of species and the speciation mechanism in islands.

Along with this review report, I provide an edited copy of the Word document, in which I added the comments and did several corrections. If you don’t receive it, please contact me (contact info at the bottom).

Experimental design

1) Molecular and taxon sampling design. This is definitely the most impressive part of the study. Assembling a molecular matrix for 186 taxa and three genes of the group in this region is a challenge for several reasons. I congratulate the authors for this job.
You must clarify how many specimens this sampling represents per area/island, both in the Results and Methods sections.

2) Phylogenetic and dating analyses. The analyses are overall good although I have several concerns. First of all, the partitioning scheme will be better assessed using PartitionFinder, than JModelTest. I don’t suggest to redo everything from scratch, I know how long it would be, but I advice the authors to run a kind of sensitivity analysis using the codon partitions and the best scheme found by PartitionFinder. Then just compare to your tree and do an Appendix.
A comment for MrBayes analyses, I think MrBayes analyses are more powerful when using the reversible-jump MCMC (rj-MCMC) for the substitution models instead of using the inferred models (by PartitionFinder) (Huelsenbeck et al. 2004 – Mol. Biol. Evol.; Bukontaite et al. 2014 – BMC Evol. Biol.). The rj-MCMC allows exploring the entire range of substitution models instead of fixing one model per partition. It is a bit longer to run for big dataset, but it is worth trying. Also I have just tried the new IQ-TREE (Nguyen et al. 2015 – MBE), a ML program that seems very promising. It is both faster and more accurate than RAxML or PhyML (probably also than GARLI), and includes a very nice algorithm that guarantees the search of the ML solution (by comparing all local optimums). In my few attempts, it’s even better than MrBayes. Finally the ultrafast bootstrap option (Minh et al. 2013 – MBE) is a small revolution in our field.

The details about the BEAST analyses are very limited. Few sentences are a bit too simple given how complex is the program and the priors you can set. Please detail more your analysis procedure. But same, there is little information about the prior settings (except for the two fossil calibrations), the burn-in, and construction of the maximum clade credibility tree for instance. I know BEAST is a very common program now, but I don’t think we need to skip the details of your analysis. After all, if one wants to reproduce your tree. How can he proceed without more information? Sorry to be insistent and rude, but this does not impede the publication of your nice work.

I am often concerned about using a particular distribution for dating a tree with fossils (or other prior information). I like the use of exponential distribution for the fossils as recommended by Ho & Phillips (2009 – Syst. Biol.). I would recommend using the uniform distribution for a sensitivity analysis. Checking the posterior distribution of the uniform distribution of the calibrated nodes may inform you on the potential problem and behaviour of your dataset. I have experimented this problem. In this respect I always prefer having a large HPD but that may encapsulate the true age, than a short but wrong HPD that can be off the true age. Similarly, I don’t recommend starting from scratch but doing a sensitivity analysis of your prior distribution on the node calibrations will be great. You can still present your dated tree with exponential distribution in the main text.

A last point concerning the dating analyses. I have personally found a drastic difference between the Yule and birth-death prior for the branching process (Condamine et al. 2015 – BMC Evol. Biol.). I am quite sure there is no effect (or little) on recently diversifying clades, but there is no simulation or still few empirical studies comparing both priors (see Toussaint et al. 2015 – Syst. Biol. who found no effect). But for more ancient clades, it could be great to have more papers doing the test in order to accumulate more evidence for selecting one of the priors. This pattern may be taxon specific, or only for ancient clades that have experienced extinction periods, but who knows. For butterflies, I have found that the Yule prior fits better than the birth-death, but with no age differences (Condamine et al. 2015 – Scientific Reports), but it was applied to a younger clade than the spiders here. That’s why it could be great to have your nice dataset analysed with both priors too. It would also be great to analyse the dataset with a coalescent tree prior since the sampling comprises intra-specific events.

3) Biogeographic analyses. The analyses and methods are fine, but DEC has the great advantage to use connectivity matrices and dispersal constraints to take into account the tectonic evolution over time. Nothing is described, which makes me think that the use of matrices has not been followed as usually done for this approach. It can have a very important impact, especially the use of the connectivity matrix (Chacon & Renner 2014 – J. Biogeogr.). You need to justify why you have set these matrices in your analyses.
Finally it has recently been shown that taking into account the founder-event as a speciation process in biogeographic analyses better fit the history of island clades (Matzke 2014 – Syst. Biol.). Although I have many concerns with DEC-J and the program BioGeoBEARS, I think it will be very interesting to analyse your dataset with this approach.

4) Figures and Tables. Overall these are very nice figures. However, no table reports the estimated ages and the 95% HPD for the main nodes (could be here for the genus crown age, and the age of the 16 putative species). This may be used as secondary calibration points for other studies.

Validity of the findings

From what is described in the text, and the analyses done, the data are robust, statistically sound, and controlled. The authors have made available their data (molecular matrix), and have deposited the DNA sequences to GenBank.

The results are valid, pending further validations with new analyses as described above. The conclusions are not overstates, and appropriate given their finding.

Few speculation is present but is not clearly stated as it. I have highlighted this part in the edited Word document (provided).

·

Basic reporting

Basically fine

Experimental design

Not really qualified to comment on the technicalities

Validity of the findings

Seems OK.

Additional comments

See attached file. Some minor issues could be addressed.

·

Basic reporting

Nice study that shows how supposedly widespread taxa actually consist of several, geographically isolated, independent evolutionary lineages, calling into question preconceived views on island biodiversity and dispersal ability. The paper is well written and illustrated

Experimental design

The methodology is adequate and arguments are for the most part convincing. I have some minor issues with the species delimitation and the time estimation but I do not think they invalidate any of the conclusions. I just would like to see some additional information

Validity of the findings

As stated above most of the conclusions reached by the authors are convincingly supported by the data and the analyses conducted

Additional comments

Some minor comments and errors I found reading the ms:
• In the mat&met, the citation provided for the 16S primers should be: Simon, C., et al. (1994). "Evolution, weighting and phylogenetic utility of mitochondrial gene sequences and a compilation of conserved polymerase chain reaction primers." Ann Entomol Soc Am 87: 651 - 701.
• Please, Justify the use of a single partition in the time estimation analysis
• Fossils provide evidence for minimum age of lineages while oceanic islands provide maximum ages for divergences between species pairs in a younger and an older island (but see Emerson, B. C. (2007). "Alarm Bells for the Molecular Clock? No Support for Ho et al.'s Model of Time-Dependent Molecular Rate Estimates." Systematic Biology 56: 337-345). However, in continental islands the situation is a little bit more complicated since species may be both the result of dispersal and or vicariance The calibration point on the Hispaniola lineage based on an amber fossil provides a minimum for the Hispaniola lineage but what is the justification for the (95%) maximum limit?
• Along the same lines, in the discussion the authors give some explanations regarding some of the time estimates that turned out to be older that the islands where the lineages were found. The only explanation provided is the existence of former islands. However a much simpler explanation would be that the author’s estimates are inaccurate. The estimated substitution rates could shed some light in the possibility of divergence time overestimation. We do have a relatively good collection of estimates for these genes in the literature and the information on the estimates substitution rates of the mtDNA could hint to some problems or consequences of the calibration points used.
• My second concern has to do with the species delimitation. The authors define a set of 22 putative lineages based on the” barcoding analysis of the COI data using MEGA6 (Table 1)”. I think there is some information lacking here. I have not been able to find any reference to MEGA distances in Table 1. Also what kind of distances was measured? (p-distance?). Regardless of the distance and the use of MEGA, I cannot find any rationale for the original split in 22 lineages. Table one provides only additional indexes on the monophyly and exclusivity of these lineages (ie. Validation) but not about their definition (ie discovery). Please elaborate a little bit more on how the initial 22 were defined.
• Did you calculate the GSI for all the genes or the COI only? It’d be goo to have these indices for the ITS as well to see how well they support the mostly mtDNA defined lineages
• I’d suggest the authors to include the Bayesian PP supports in Fig 1, is an easy way to show congruence across different methods
• In the results, it’d be informative to specify that the clades discusses were or not supported (ie MLboot>75 or any other threshold you may justify and/or PP>0.95). For example “specimens from the Yucatan peninsula are sister to the islands clade” but this clade is not supported in a statistical sense (low BS and PP)
• There are several references in the text to unpublished data under different categories (in prep, submit, in review). I guess that for the purposes of this publication these are all unpublished data.

---

## Round 0.2 · accepted · Accept

Dear Authors,

Congratulations for the powerful modifications made since the first version. On this basis, I am glad to recommand that your manuscript is accepted for publication in PeerJ.

Thanks again for submitting your manuscript,

P. Grandcolas

·

Basic reporting

I am reviewer #1 (Fabien Condamine).
I have been quite 'harsh' during my first round of review, but it was in the aim to improve the study. I think the authors have accomplished a nice revision of their study, which satisfies me. I am overall supportive of publishing this paper in PeerJ.

Experimental design

The authors have realized new analyses to support their findings. I am happy with this effort, and I think it makes the study stronger and more thorough.
I understand that the dataset is 'fairly simple' but you are the owner, the readers are not, so one can rise doubts on the results, especially the dating analyses.
Now, with the new results, I am more convinced the study is robust to some biases.

Validity of the findings

After reading this revision, I found the results stronger than before. I think the paper is worth being published given the topic it addresses. I think this situation occurs in many cases, such that I hope this paper will inspire further studies in other model system.

Additional comments

Congratulations on the good job done so far.

·

Basic reporting

In my former review I already considered the paper to be acceptable for publication once minor changes were incorporated. I have now read the rebuttal letter of the authors and I am happy with all the modifications they have introduced and the way they have delt with my comments and suggestions. Therefore I have no further comment on the ms and cpnsier it ready for publication.

Experimental design

see above

Validity of the findings

see above

Additional comments

Nice work